# Perioperative changes of response to antiplatelet medication in vascular surgery patients

Thomas Hummel[1]*, Saskia Hannah Meves[2¤a], Andreas Breuer-Kaiser[3], Jan-Ole Düsterwald[1¤b], Dominic Mühlberger[1], Achim Mumme[1], Horst Neubauer[4¤c]

**1** Department of Vascular Surgery, St. Josef Hospital, Katholisches Klinikum Bochum, Ruhr University Bochum, Bochum, Germany, **2** Department of Neurology, St. Josef Hospital, Katholisches Klinikum Bochum, Ruhr University Bochum, Bochum, Germany, **3** Department of Anaesthesiology, St. Josef Hospital, Katholisches Klinikum Bochum, Ruhr University Bochum, Bochum, Germany, **4** Department of Cardiology, St. Josef Hospital, Katholisches Klinikum Bochum, Ruhr University Bochum, Bochum, Germany

¤a Current address: Department of Neurology, St. Marien-Hospital, Mülheim an der Ruhr, Germany
¤b Current address: Department of Vascular Surgery, Bethanien Hospital, Moers, Germany
¤c Current address: Department of Cardiology, St. Elisabeth-Hospital, Herten, Germany
* thomas.hummel@rub.de

**Data Availability Statement:** All relevant data are within the paper and its Supporting Information files.

## Abstract

### Introduction

Reduced antiplatelet activity of aspirin (ALR) or clopidogrel (CLR) is associated with an increased risk of thromboembolic events. The reported prevalence data for low-responders vary widely and there have been few investigations in vascular surgery patients even though they are at high risk for thromb-embolic complications. The aim of this prospective observational monocentric study was to elucidate possible changes in ALR or CLR after common vascular procedures.

### Methods

Activity of aspirin and clopidogrel was measured by impedance aggregometry using a multiple electrode aggregometer (Multiplate®). Possible risk factors for ALR or CLR were identified by demographic, clinical data and laboratory parameters. In addition, a follow-up aggregometry was performed after completion of the vascular procedure to identify changes in antiplatelet response.

### Results

A total of 176 patients taking antiplatelet medications aspirin and/or clopidogrel with peripheral artery disease (PAD) and/or carotid stenosis (CS) were included in the study. The prevalence of ALR was 13.1% and the prevalence of CLR was 32% in the aggregometry before vascular treatment. Potential risk factors identified in the aspirin group were concomitant insulin medication (p = 0.0006) and elevated C-reactive protein (CRP) (p = 0.0021). The overall ALR increased significantly postoperatively to 27.5% (p = 0.0006); however, there

**Funding:** The authors received no specific funding for this work.

**Competing interests:** The authors have declared that no competing interests exist.

was no significant change in CLR that was detected. In a subgroup analysis elevation of the platelet count was associated with a post-procedure increase of ALR incidence.

## Conclusion

The incidence of ALR in vascular surgery patients increases after vascular procedures. An elevated platelet count was detected as a risk factor. Further studies are necessary to analyse this potential influence on patency rates of vascular reconstructions.

## Introduction

Antiplatelet drugs are the most frequently used medications for prevention of coronary and peripheral occlusive disease events. In secondary and tertiary prevention, aspirin or clopidogrel is recommended by the professional associations to prevent thromboembolic events (European guidelines for PAD and carotid stenosis) [1]. However, primarily performed studies in cardiological patients have shown that there exist individual variations in the effectiveness of the antiplatelet medication [2–4]. The relevance of an effective antiplatelet therapy in vascular surgery patients could be shown in surveillance studies, where an improvement in patency rates after peripheral revascularization of the lower extremity was demonstrated [5]. In addition, there was an increased rate of major adverse limb events shown in patients with a reduced antiplatelet effect of the platelet aggregation inhibitor after a peripheral endovascular therapy [6]. The prevalence of treatment failures varies and vascular surgery patients are still underrepresented, even though they have a high risk for thromb-embolic events. In an initial prevalence study in patients undergoing common vascular procedures, the prevalence of high on-treatment platelet reactivity (HPR, low response (LR)) was 19.3% for aspirin (ALR) and 21.1% for clopidogrel (CLR) measured by impedance aggregometry [7].

The present study was initiated to investigate changes of the response status of the antiplatelet prophylaxis in vascular surgery patients with symptomatic PAD or internal carotid stenosis after common vascular procedures. We hypothesized that low response is increased after common vascular procedures, and more invasive procedures will lead to a higher incidence of low response rate.

## Materials and methods

This empiric, monocentric study was reviewed and approved by the responsible ethics committee of the Ruhr University Bochum and was conducted in accordance with national law and the Declaration of Helsinki of 1975 (as currently amended). All patients were included by written informed consent. The recruitment of the participants at the time of inpatient admission took place at the vascular surgery department of the St. Josef Hospital, university hospital of the Ruhr-University of Bochum from February 2010 to December 2011. The prospective observational study was part of a graduation as a medical doctor of the Ruhr-University of Bochum.

We included vascular surgery patients with symptomatic PAD and/or an internal carotid stenosis with an on-going antiplatelet medication with aspirin, clopidogrel or a dual medication of both drugs, requiring a common vascular procedure in the study. The performed procedures were divided into five different invasive groups: Diagnostic angiography, interventional treatments using PTA and stenting of the pelvic and leg arteries, peripheral

bypass operations of the lower extremity, TEAs of the inguinal arteries and TEAs of the carotid artery. There were no hybrid therapies performed in this investigation, in order to achieve consistent results. Demographic data of the patients, the reason for admission and the different procedures are presented in Table 1.

In order to rule out non-adherence or irregularities in the ongoing medication, patients were questioned to proclaim whether they had taken their antithrombotic medication daily at the same time during the last 14 days before they were included in the study. In addition to the demographic data, we documented concomitant diseases, concomitant medication and laboratory parameters in order to detect any risk factors for a low response (LR) to the antiplatelet medication (high on-treatment platelet reactivity (HPR)). In order to assess the severity of the atherosclerotic vascular disease, we documented the location and the severity of the PAD according to the Fontaine classification.

Due to the design as a observational study the exclusion criteria were limited to general contraindications to participation in a scientific study as age, cognitive disorders, pregnancy. Further exclusion criteria were contraindications to the study medication and extrinsic or intrinsic activation / or depression of platelet activity (Gastrointestinal ulceration with history of bleeding within six weeks, NYHA stage 4, kidney or liver failure, abnormal platelet count, known coagulation disorders, cancer, ASA or Clopidogrel loading dose within the last 14 days).

After inclusion to the study, blood samples were collected from each patient in the morning of the day before the intervention from a cubital vein with a 21-gauge needle and at the same time in the morning of the day after the vascular procedure. The plunger of the Monovette tube was pulled back gently and steadily in order to prevent shear stress on the platelets. The first 4 ml of whole blood were discarded in order to prevent spontaneous activation of the platelets. For the assays, 4-ml tubes with sodium citrate and 2.7-ml tubes with r-hirudine as anticoagulant were used. Because of the refractory period of the platelets after blood sampling [8, 9], blood samples were stored for at least 30 minutes at room temperature before processing. All blood analyses were performed within two hours after sample collection, as in this time span the results are most constant [10].

In order to assess the antiplatelet effect of aspirin and clopidogrel, we performed whole blood aggregometry using a multiple electrode impedance aggregometer (Multiplate®

**Table 1. Baseline demographic, clinical and treatment data.**

| Total | N = 176 |
|---|---|
| Mean age in years (SD) | 74 (±10) |
| Male (%) | 61 (34.7) |
| Female (%) | 115 (65.3) |
| Carotid-Stenosis (%) | 74 (42.1) |
| PAD (%) | 102 (58.0) |
| **Treatment** | **Number** |
| Stenting/PTA (%) | 47 (26.7) |
| DSA (%) | 19 (10.8) |
| Peripheral bypass surgery (%) | 25 (14.2) |
| Femoral TEA (%) | 17 (9.7) |
| Carotid TEA (%) | 68 (38.6) |

(PAD: peripheral artery disease, PTA: percutaneous transluminal angioplasty, DSA: digital subtraction angiography, TEA: thromb-endarteriectomy).

Analyzer, Roche, Switzerland) in which measurements are performed with two pairs of measuring electrodes simultaneously. A deviation of individual results of more than 20% from the mean or a correlation coefficient between the two measurements below 0.98 was interpreted as error and the measurement repeated. The results were interpreted according to the manufacturer's instructions and the methods for aspirin and clopidogrel already described in literature [11–14]. For analysis, the area under the aggregation curve (AUC) is used. The change in the impedance between the electrodes is given in the unit AU (arbitrary aggregation unit). The area under the aggregation curve has the unit AU*min. As this usually gives a three-digit figure, the unit U (Unit) is used, where 1 U = 10 AU*min. The measurements of the aggregation capacity on aspirin were performed using the ASPI solution. This causes platelet activation by arachidonic acid, which is transformed to thromboxane A2 via cyclooxygenase (COX) together with thromboxane synthase. The measurement of the aggregation capacity on clopidogrel was performed with ADP solution for quantitative in-vitro determination of the platelet function after stimulation of the adenosine diphosphate (ADP) receptor of the platelets. In accordance with the manufacturer's instructions and previous investigations a cut-off value of $> 46$ U was used for ALR and of $> 40$ U for CLR [4, 15].

Before initiating the study a sample size planning was performed using a significance level of $\alpha = 0.05$, estimated Pearson´s correlation coefficient of r = 0,25 and stochastic power of 0,8. The sample size of the study calculated in this way was n = 123 in order to get a significant result. Statistical analysis was performed using Microsoft Excel (Version 14.6.4, 2010, Microsoft Corporation) and GraphPad Prism 7 software (GraphPad Software Inc. 2016) (causes of low response was performed with students t-test for independent samples and the chi-squared test, the comparison of pre- and post-procedure response with McNemar test). Data are presented as frequency distributions and percentages. $P < 0.05$ was regarded significant. For post hoc adjustments for multiple comparisons in the risk factor analysis we used the Holm-Bonferroni method.

## Results

A total of 176 patients were included in the study. The mean age of the participants was 74 years (SD10), 61 (34.7%) were male and 115 (65.3%) were female. 74 (42.1%) patients were hospitalised for treatment of a carotid stenosis and 102 (58.0%) due to a symptomatic PAD (see Table 1). Altogether, there were 148 (84.1%) patients on a continuous medication with 100 mg aspirin, twelve (6.8%) on dual antiplatelet medication with 100 mg aspirin and 75 mg clopidogrel, eleven (6.3%) on monotherapy with 75 mg clopidogrel, two (1.1%) on dual antiplatelet medication with 300 mg aspirin and 75 mg clopidogrel and three (1.7%) on monotherapy with 300 mg aspirin.

### Pre-procedure response status

Inadequate antiplatelet activity was found in 21 (13.1%) of the 160 patients taking 100 mg/d aspirin and in one (20%) of the five patients taking 300 mg/d aspirin. Among the 25 patients taking 75 mg/d clopidogrel, there were eight (32%) who showed a low response (Fig 1).

Due to the low numbers of cases in the other groups, the analysis of the risk factors for pre-procedure low response was only performed in the patients taking 100 mg aspirin. Both, the demographic data and concomitant medication and laboratory parameters were documented and analysed as possible risk factors for low response to the antiplatelet medication. For this, the responders and low responders were compared for significant differences.

The comparison revealed no significant differences within the demographic data. In the analysis of concomitant diseases and co-medication, patients with diabetes mellitus (p = 0.049)

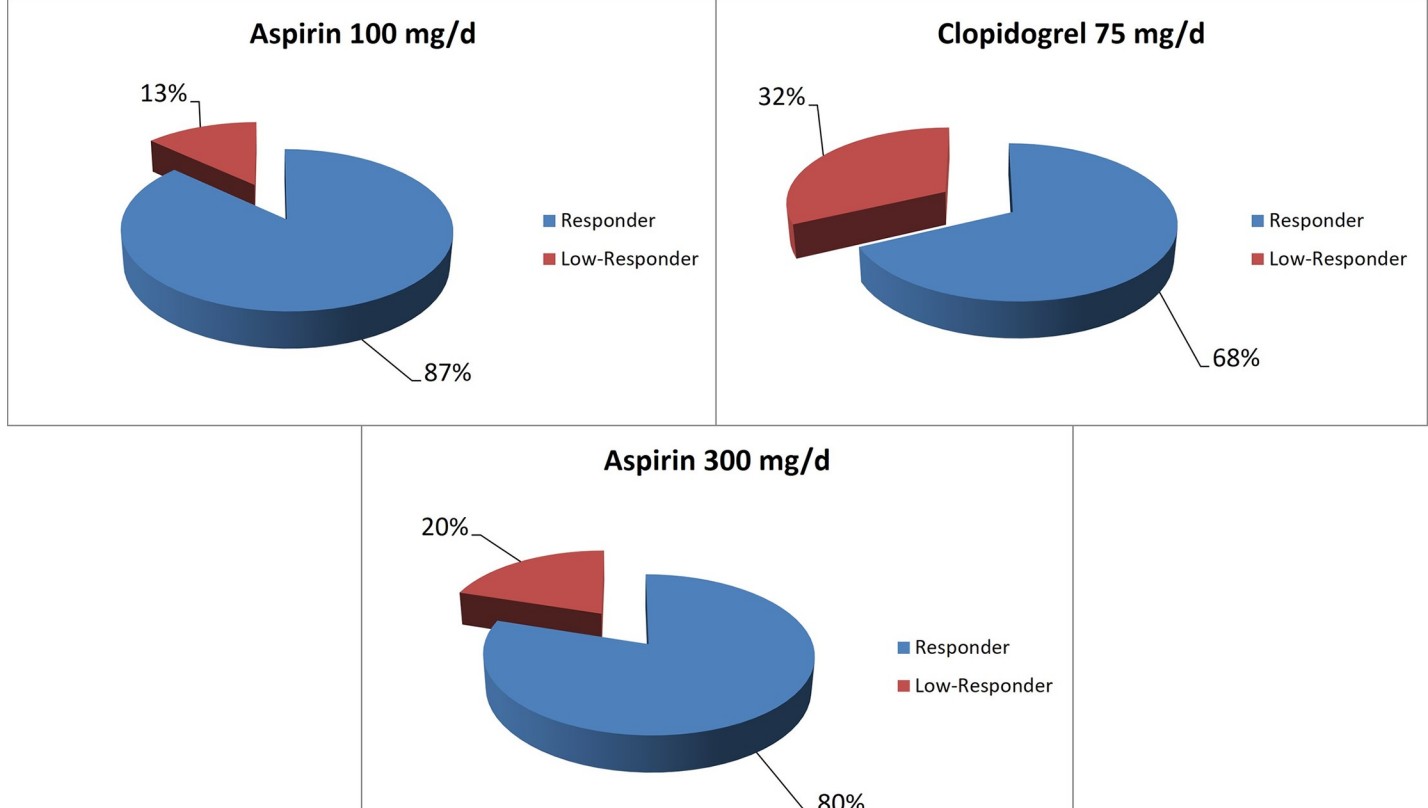

**Fig 1. Preprocedural prevalence of low response.** The pie charts show the preprocedural proportion in percentage of low responders detected by multiple electrode impedance aggregometer in the different medication groups.

and concomitant medication with insulin (p = 0.0006) showed a increased LR rate. In addition, patients with pre-procedure low response showed a elevated CRP (p = 0.0021) (Table 2). However, the post hoc test for multiple comparisons using the Holm-Bonferroni method showed the increased LR rate in patients with diabetes mellitus as no significant (adjusted significance level p = 0.0083), but confirmed the significant changes of LR in concomitant medication with insulin (adjusted significance level p = 0.0038) and elevated CRP (adjustet significance level p = 0.0056).

## Post-procedure changes in response

In the group of patients with 300 mg aspirin and 75 mg clopidogrel the pre- and post-procedure comparison showed no significant differences. Due to the small number of cases and the consecutive limitation in the interpretation of these data, the further postprocedural analysis was performed exclusively in the group taking 100 mg aspirin.

In this group the pre-/post-procedure comparison showed a significant increased frequency of a low response from 21 (13.1%) to 44 (27.5%) (p = 0,0006) (Fig 2).

107 (66,9%) of the 160 patients treated with 100 mg/d aspirin stayed responders pre- and post-procedure, 12 (7,5%) patients stayed periprocedural low responders and 32 (20%) patients switched from pre-procedure response to post-procedure low response. 9 (5,6%) pre-

**Table 2. Preprocedural risk analysis for aspirin low response (100 mg/d).**

| | Responder | Low-Responder | p-value |
|---|---|---|---|
| N = 160 (%) | 139 (86.88) | 21 (13.13) | |
| Average age in years (SD) | 74 (±10) | 76 (±10) | 0.38 |
| Male (%) | 53 (38.1) | 6 (28.6) | 0.40 |
| BMI > 30 kg/m$^2$ (%) | 28 (20.1) | 6 (28.6) | 0.38 |
| **Concomitant disease** | | | |
| Arterial hypertension(%) | 104 (74.8) | 15 (71.4) | 0.74 |
| Diabetes mellitus (%) | 37 (26.6) | 10 (47.6) | 0.049 |
| Smoking (%) | 95 (68.3) | 16 (76.2) | 0.47 |
| Hypercholesterinämia (%) | 36 (25.9) | 4 (19.1) | 0.50 |
| Prior stroke (%) | 16 (11.5) | 2 (9.5) | 0.33 |
| CHD (%) | 46 (33.1) | 6 (28.6) | 0.68 |
| **Concomitant medication** | | | |
| ACE-inhibitors (%) | 83 (59.7) | 15 (71.4) | 0.30 |
| Beta-blockers (%) | 70 (50.4) | 11 (52.4) | 0.86 |
| Ca-channel blockers (%) | 51 (36.7) | 8 (38.1) | 0.90 |
| Diuretics (%) | 54 (38.9) | 12 (57.1) | 0.11 |
| Oral antidiabetics (%) | 19 (13.7) | 2 (9.5) | 0.60 |
| Insulin (%) | 11 (7.9) | 7 (33.3) | **0.0006** |
| Nitrats (%) | 20 (14.4) | 2 (9.5) | 0.55 |
| Statins (%) | 79 (56.9) | 9 (42.9) | 0.23 |
| Proton pump inhibitors (%) | 33 (23.7) | 4 (19.1) | 0.63 |
| Pantoprazol (%) | 19 (13.7) | 3 (14.3) | 0.94 |
| Omeprazol/Esomeprazol (%) | 13 (9.4) | 1 (4.8) | 0.49 |
| Antidepressivs (%) | 8 (5.8) | 2 (9.5) | 0.50 |
| Number of co-medication (SD) | 4.87 (±3.16) | 6.10 (±3.13) | 0.10 |
| **Underlying disease** | | | |
| Carotid stenosis (%) | 60 (43.2) | 5 (23.8) | 0.09 |
| PAD (%) | 79 (56.8) | 16 (76.2) | 0.09 |
| **Laboratory parameters** | | | |
| Leucozytes (x10$^9$/l) (SD) | 8099 (±2157) | 8281 (±1632) | 0.71 |
| Hämoglobin (g/dl) (SD) | 13.76 (±1.70) | 13.74 (±1.78) | 0.96 |
| Platelet count (x10$^9$/l) (SD) | 236,87 (±83,71) neu | 247.8 (±63.00) | 0.57 |
| Serum creatinine (μmol/l) (SD) | 1.15 (±0.59) | 1.30 (±1.12) | 0.35 |
| GFR (ml/min) (SD) | 68.57 (±31.95) | 67.22 (±34.34) | 0.86 |
| HbA$_{1c}$ (%) (SD) | 6.0 (±0.86) | 6.30 (±1.16) | 0.16 |
| Total cholesterol (SD) | 203.11 (±40.84) | 201.33 (±32.22) | 0.85 |
| LDL/HDL cholesterol (SD) | 2.33 (±0.98) | 2.59 (±1.10) | 0.27 |
| CRP (mg/l) (SD) | 4.64 (±10.71) | 16.90 (±37.86) | **0.0021** |

Data presented are numbers, mean ± SD or percentage.

(BMI: body mass index, CHD: coronary heart disease, ACE: angiotensin-converting enzyme, Ca: calcium, GFR: glomerular filtration rate, PAD: peripheral artery disease, PTA: percutaneous transluminal angioplasty).

procedure low responders demonstrated post-procedure response. Each of these cases was induced by an intraprocedural loading dose with 500 mg aspirin.

In order to detect possible risk factors for aspirin low response as a result of the vascular treatment, we performed an analysis within the most informative group of the pre-procedure

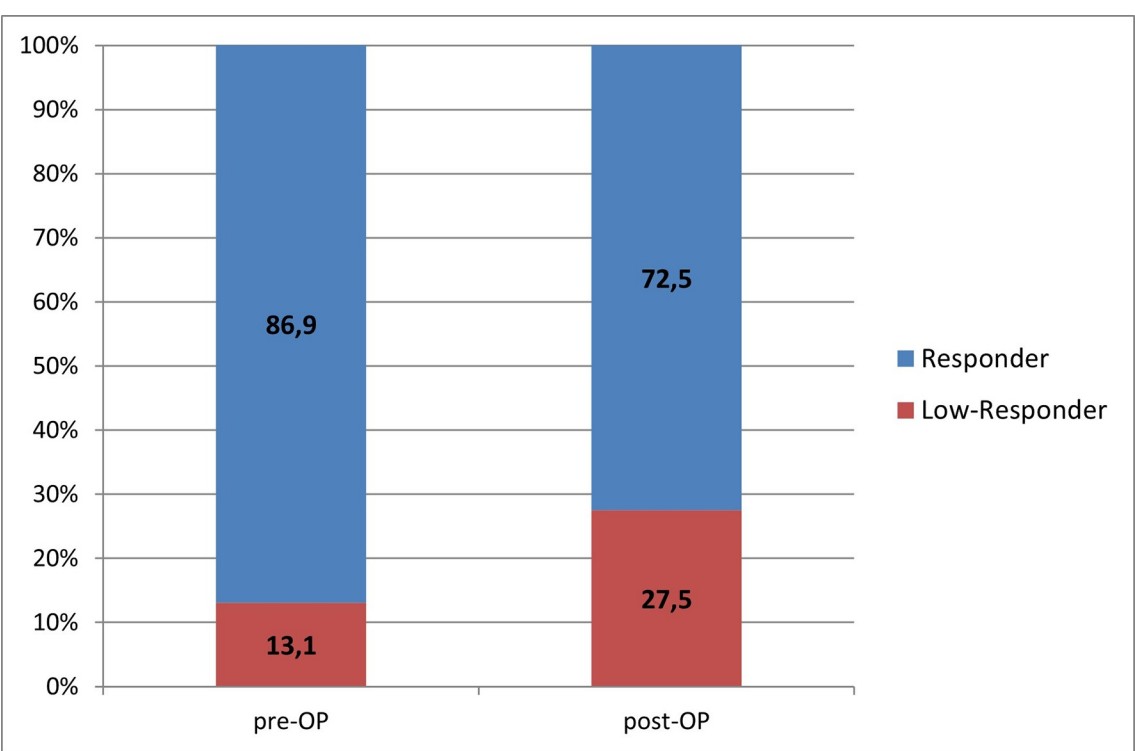

**Fig 2. Pre- und postprocedural response (ALR) with aspirin 100 mg treatment.** The bar chart shows the periprocedural change in response to aspirin 100 mg/d medication in percentage detected by multiple electrode impedance aggregometer.

responders treated with 100 mg aspirin (n = 139). These 107 (77.%) post-procedure responders were compared with the 32 (23.%) patients that switched to post-procedure low response. Overall, no significant differences were found regarding demographic data, concomitant diseases or concomitant medication. The reasons for hospital admission and the severity of the PAD according to Fontaine did not differ significantly. 15 (46.9%) patients with a carotid TEA, 5 (15.6%) patients with femoral TEA and 7 (21.9%) patients with a peripheral bypass changes response status to ALR, while 2 (6.3%) patients with diagnostic angiography and 3 (9.4%) patients with an interventional treatment were postprocedural ALR. Regarding the different invasiveness of the performed vascular procedure, the overall group of patients undergoing open vascular surgery with carotid revascularisation, TEA of the femoral artery and peripheral bypass of the lower extremity showed an increase in low response (p = 0,0104). In post hoc analysis with Holm- Bonferroni method this finding is changing to being not significant with a adjusted significance level of p = 0.0083. The analysis of the individual operations performed showed no significant differences. A minimal invasive vascular intervention (PTA/ Stenting) was found less often amongst patients with a post-procedure low response (p = 0.015). This was not statistical significant in a post hoc analysis (adjusted significance level of p = 0,01) In the examination of the post-procedure laboratory values, significant differences for the development of periprocedural low response were found for a higher platelet count (p = 0.0008, post hoc adjusted significance level p = 0.0056). After post hoc adjustment a lower creatinine value was not significant associated with postprocedural LR (p = 0.0403, adjusted significance level p = 0.0063). All other laboratory parameters showed no influence on the post-procedure response status (Table 3).

**Table 3. Postprocedural risk analysis for the development of aspirin low response (100 mg) comparing responder and low-responder.**

| | Postprocedural Responder | Postprocedural Low-Responder | p-value |
|---|---|---|---|
| N = 139 (%) | 107 (77.0) | 32 (23.0) | |
| Average age in years (SD) | 74 (±10) | 73.40 (±12) | 0.89 |
| Male gender (%) | 37 (34.6) | 16 (50.0) | 0.12 |
| BMI > 30 kg/m$^2$ (%) | 22 (20.6) | 6 (18.8) | 0.82 |
| **Concomitant disease** | | | |
| Arterial hypertonsion (%) | 83 (77.6) | 21 (65.6) | 0.17 |
| Diabetes mellitus (%) | 30 (28.0) | 7 (21.9) | 0.49 |
| Smoking (%) | 76 (71.0) | 19 (59.4) | 0.21 |
| Hypercholesterinämia (%) | 31 (29.0) | 5 (15.6) | 0.13 |
| Prior stroke (%) | 12 (11.2) | 4 (12.5) | 0.84 |
| CHD (%) | 39 (36.4) | 7 (21.9) | 0.12 |
| **Concomitant medication** | | | |
| ACE inhibitors (%) | 65 (60.8) | 18 (56.3) | 0.65 |
| Beta-blockers (%) | 57 (53.3) | 13 (40.6) | 0.21 |
| Ca-chanal blockers (%) | 42 (39.3) | 9 (28.1) | 0.25 |
| Diuretics (%) | 45 (42.1) | 9 (28.1) | 0.16 |
| Oral antidiabetiks (%) | 15 (14.0) | 4 (12.5) | 0.83 |
| Insulin (%) | 10 (9.4) | 1 (3.1) | 0.25 |
| Nitrats (%) | 17 (15.9) | 3 (9.4) | 0.36 |
| Statins (%) | 62 (57.9) | 17 (53.1) | 0.63 |
| PPI (%) | 26 (24.3) | 7 (21.9) | 0.78 |
| Pantoprazol (%) | 14 (13.1) | 5 (15.6) | 0.71 |
| Omeprazol/Esomeprazol (%) | 12 (11.2) | 1 (3.1) | 0.17 |
| Antidepressivs (%) | 6 (5.6) | 2 (6.3) | 0.89 |
| Number of co-medication (SD) | 4.87 (±3.13) | 4.87 (±3.26) | 1.00 |
| **Underlying disease** | | | |
| Carotid stenosis (%) | 44 (41.1) | 16 (50.0) | 0.37 |
| PAD (%) | 63 (58.9) | 16 (50.0) | 0.37 |
| **Treatment** | | | |
| Peripheral bypass (%) | 15 (14.0) | 7 (21.9) | 0.29 |
| Femoralic-TEA (%) | 9 (8.4) | 5 (15.6) | 0.23 |
| Diagnostic angiography (%) | 10 (9.4) | 2 (6.3) | 0.58 |
| PTA/Stenting (%) | 33 (30.8) | 3 (9.4) | 0.015 |
| Carotid-TEA (%) | 40 (37.4) | 15 (47.0) | 0.34 |
| Open operative approach (%) | 64 (59.8) | 27 (84.4) | 0.0104 |
| **PAD-Classification (Fontaine)** | | | |
| Stage 1 (%) | 3 (4.5) | 0 | 0.34 |
| Stage 2 (%) | 45 (67.2) | 14 (73.7) | 0.86 |
| Stage 3 (%) | 5 (7.5) | 2 (10.5) | 0.72 |
| Stage 4 (%) | 14 (21.0) | 3 (15.8) | 0.57 |
| **Laboratory parameters** | | | |
| Leucozytes (x10$^9$/l) (SD) | 7949 (±2066) | 8616 (±2376) | 0.12 |
| Hämoglobin (g/dl) (SD) | 13.88 (±1.50) | 13.36 (±2.23) | 0.13 |
| Platelet count (x10$^9$/l) (SD) | 224.4 (±71.10) | 280.0 (±106.60) | **0.0008** |
| Serum-creatinin (µmol/l) (SD) | 1.21 (±0.64) | 0.97 (±0.25) | 0.0403 |
| GFR (ml/min) (SD) | 66.24 (±32.52) | 76.80 (±28.34) | 0.10 |
| HbA$_{1c}$ (%)(SD) | 6.04 (±0.82) | 5.88 (±0.99) | 0.36 |

(*Continued*)

**Table 3.** (Continued)

|  | Postprocedural Responder | Postprocedural Low-Responder | p-value |
|---|---|---|---|
| Total cholesterol (SD) | 203.40 (±40.29) | 202.0 (±42.83) | 0.87 |
| LDL/HDL cholesterol (SD) | 116.02 (±1.00) | 120.23 (±39.12) | 0.26 |
| CRP (mg/l) (SD) | 4.52 (±11.20) | 5.12 (±8.49) | 0.78 |

Data presented are numbers, mean ± SD or percentage.

BMI = body mass index, CHD = coronary heart disease, ACE = angiotensin-converting enzyme, Ca = calcium, PPI = proton pump inhibitor, PAD = peripheral artery disease, PTA = percutaneous transluminal angioplasty, TEA = thromb-endarteriectomy, GFR = glomerular filtration rate.

## Discussion

The prevalence of a low response to 100 mg aspirin (ALR) in patients with symptomatic PAD and/ or an internal carotid stenosis was 13.1% in this study. The prevalence of a low response to 75 mg clopidogrel (CLR) was 32%. These patients were already without measurable prophylaxis against thromboembolic events before the necessary vascular intervention.

Al-Azzam et al. investigated 418 patients on antiplatelet medication. They found a prevalence of ALR of 18.7% using the Multiplate® aggregometer [4]. In our previous monocentric prevalence study in vascular surgery patients using impedance aggregometry (ChronoLog® 590), the prevalence of ALR was 19.3% and that of CLR 21.1% [7]. In a meta-analyses, the mean prevalence of ALR in cardiological patients was 27.1%, and that of CLR 21% [3, 16]. Thus, compared with cardiological patients, the prevalence of ALR in our study was lower and the detected prevalence of CLR was higher, but it must be admitted that the results regarding CLR in our study are generally limited, as the number of cases in the clopidogrel group was small.

However, it is important to note that there are various established laboratory tests for the detection of reduced activity of aspirin and clopidogrel. Therefore the comparability of the values is limited [17]. This could also explain the relatively wide range of the prevalence of low response in the literature.

In our study, co-medication with insulin and an elevated CRP were verified as risk factors for pre-procedure ALR. In a study with 358 patients with an acute coronary syndrome, Modica et al. were able to show that an elevated CRP was an independent predictor of ALR [18]. Inflammatory processes have also been detected as risk factors for ALR [19, 20]. Furthermore, platelet-leucocyte interactions are described in the thrombotic milieu of acute cerebral ischemia [21]. Other studies also show a positive correlation of both, the platelet and the leukocyte count, with platelet aggregation [22, 23]. In our analysis, we did find a higher leukocyte count in the low responders compared with the responder group but the difference was not statistically significant.

Risk factor analyses for a low-response status in meta-analyses and systematic reviews were mainly examined in cardiological patients. In confirmation with our findings, insulin-dependent diabetes mellitus has been found to be a risk factor for ALR as well. In addition, an increased risk for ALR and/or CLR was found in patients with hypercholesterolemia, elevated body mass index, acute coronary syndrome, active nicotine consumption, medication with NSAIDs (particularly ibuprofen and naproxen) and medication with proton pump inhibitors. A correlation was also found between genetic factors and hyperhomocysteinemia and a low response status [24–26].

In our study the frequency of post-procedure ALR was doubled in comparison to the pre-procedure results. The change in the response status from responder to low responder was significantly associated with the vascular treatment.

In other aggregometry studies, operative vascular surgery procedures (peripheral revascularisation, operations on the abdominal aorta and in the carotid circulation) correlated with the ALR prevalence [27, 28|. Hagedorn et al. used Multiplate® aggregometry to investigate 17 patients undergoing elective aortic surgery or a peripheral vascular operation with regard to an operation related increase in ALR. This showed an increase in ALR from 12% to 41% [29]. Rajagopalan et al. showed a significant increase in ALR after major vascular surgery using VerifyNow® aggregometry (Instrumentation Laboratory, Bedford, Massachusetts). The authors hypothesised that the cause was platelet activation induced by the operative trauma [28]. Moake et al. demonstrated platelet activation induced by vascular shear stress after vascular interventions. This is mediated by von-Willebrand-Factor (VWF) and ADP [30]. According to the theory of the induction of a ALR by the vascular trauma, the ALR rates would be expected in a direct correlation to the invasiveness of the intervention. However, we could not find any significant differences in ALR rates between the differently invasive procedures in our study.

In addition to platelet activation, as a result of the vascular procedure, an increased platelet turnover is discussed in literature as a cause of ALR. This should lead to an increased number of uninhibited platelets. Perneby et al. described, that based on a platelet renewal rate of 10% per day with a single aspirin dose/d and a half-life of approximately 15 min, there is always a certain proportion of uninhibited platelets in circulation, and that this rises in the case of an increased platelet turnover [31]. Guthikonda et al. showed that increased immature reticular platelets, which are still capable of thromboxane synthesis might counteract the antiplatelet effect of aspirin [32].

The possibility of a transient intra-/post-procedure low response is also discussed. Payne et al. examined platelet aggregation before, during and after a carotid operation in 50 patients taking 150 mg/d aspirin. A control group comprised of 18 patients undergoing PTA. In both groups there was a marked intraoperative increase in platelet aggregation, which had fallen again 4 hours after the vascular procedure [27].

In the risk factor analysis of this study elevated platelets were significant associated to a post-procedure ALR. This is supported by studies on patients with thrombocytosis, which were able to demonstrate a positive correlation between platelet count and impedance aggregation [33]. Reduced creatinine value and the invasiveness of the procedure failed to be significant in the post hoc adjustment in the risk factor analysis. However, Neubauer et al., identified an elevated creatinine level as a risk factor for ALR using whole blood aggregometry (Chronolog 590) [34].

Lack of effective antiplatelet prophylaxis can have serious consequences for the patient. Studies have shown a significant increase in clinically relevant ischemic cardiovascular events in the group of low responders [16, 35, 36]. This resulted in a fourfold increased risk for cardiovascular events in these patients (pooled odds ratio of 3.8; 95% confidence interval: 2.3–6.1) [24, 35, 37]. Zimmermann and Hohlfeld even assume a 13-fold increased risk for atherothrombotic events in ALR patients [38]. Price et al. showed a significant increase in the 6-month mortality rate in cardiological patients with CLR [39].

Whether absent or reduced antiplatelet action of aspirin or clopidogrel after successful operative revascularisation (thus with inadequate tertiary pharmacological prophylaxis) leads to a reduced patency rate of the reconstruction remains unclear. Investigations in the context of graft surveillance or outcome surveillance after surgical treatment in patients showing low response are not currently available.

The available studies on minimally invasive endovascular treatment show differing results for CLR with regard to short-term surveillance. In a retrospective study in 385 patients undergoing endovascular treatment, Bernlochner et al. were not able to show any significant difference in the outcome of the patients after one year using Multiplate® aggregometry [40].

Spiliopoulos et al., on the other hand, showed a highly significant association between a low response and the 1-year outcome of patients with PAD undergoing endovascular treatment using VerifyNow-aggregometry [41].

There are some limitations and confounding factors of our study. A methodological limitation of our research and the literature in general is that to date there is no consensus on the preferred diagnostic test in the detection of low response to antiplatelet medication and the cut-off values for ALR used for multiple electrode impedance aggregometry, so it is difficult to compare and evaluate the results of different approaches. Maybe this is one main reason for the varies results in the detection of low response to antiplatelet medication. In spite of a detailed medication history, it remains unclear to what extent the results in our study were influenced by non-adherence to the antiplatelet medication. Lack of medication adherence as the reason for a low response is reported in literature in 9% of all cases [42]. Furthermore, due to low number of cases in the different groups, the impact of the risk factors analysis is low. To identify significant risk factors for the periprocedural development of ALR, further investigations with higher number of cases are required. One major limitation of our study is the lack of long term follow up response status of antiplatelet medication. Based on the results of this investigation no conclusion can be drawn, if the detected postprocedural increased ALR is transient or persistent. However, a differentiation is required for the development of a therapeutic concept to overcome this detected LR after common vascular procedures. This could probably take place in a test-and-treat concept. In this context the first results of the prospective ADAPTABLE study (Aspirin Dosing: A Patient-centric Trial Assessing Benefits and Long-term Effectiveness) are eagerly awaited with regard to an optimal dose finding in a benefit-risk evaluation of ASA prophylaxis [43]. The doses of 81 and 325 mg aspirin available in the USA for secondary prophylaxis are compared in this trial. In 2019, patient enrollment with 15,000 subjects was completed. The first results have been announced for 2020. The "one size fits all" strategy, that has been practiced so far, may then have to be reconsidered. One further limitation of this study is the low significance of the results concerning Clopidogrel. Due to the small number of cases, no reliable conclusions could be drawn in the clopidogrel group for the perioperative course of response status and risk factors of CLR.

In addition, it is difficult to evaluate the results of our study in terms of clinical outcome. We did not perform a clinical outcome measurement to detect postprocedural thromboembolic events. Due to this, the clinical impact of our findings remains unclear. Furthermore, we could not prove any significant difference between interventional or surgical treatments, so that unfortunately no test strategy depending on the invasiveness of the procedure carried out can be derived from this examination. The currently available evidence regarding the clinical course of the vascular patients with ALR or CLR does not justify to recommend general testing of all vascular patients. Although we could show a twofold increase in the ALR rate after common vascular procedures, testing should be reserved for high-risk patients or patients with a history of postoperative thromboembolic events and conspicuous clinical course in a cost-benefit ratio [44, 45]. Further investigations with clinical endpoints and follow up platelet response measurements are needed to answer the question which patient should be tested. In our daily clinical practice patients with unexplainable graft failure, multiple revascularisation and thromboembolic ischemia are considered as high-risk and are evaluated for ALR or CLR.

## Conclusion

This study showed that the prevalence of ALR in vascular surgery patients is at the lower end of the ranges reported in literature. There was an approximate two fold increase of low responders to aspirin medication after common vascular procedures (13.1% to 27.5%)

Whether the reduced antiplatelet effect has a negative influence on the patency rate of the vascular procedures or the clinical outcome could not be answered because of unavailable long-term data. With respect to the numerous non-standardised methods for measuring platelet aggregation, the comparability of the results reported in literature is limited. Regarding the risk factors, studies with larger vascular surgery populations and uniform measuring methods are necessary.

## Supporting information

**S1 Data.**
(XLSX)

## Author Contributions

**Conceptualization:** Thomas Hummel, Saskia Hannah Meves, Andreas Breuer-Kaiser, Dominic Mühlberger, Achim Mumme, Horst Neubauer.

**Data curation:** Thomas Hummel, Saskia Hannah Meves, Andreas Breuer-Kaiser, Jan-Ole Düsterwald, Horst Neubauer.

**Investigation:** Thomas Hummel, Saskia Hannah Meves, Jan-Ole Düsterwald, Horst Neubauer.

**Supervision:** Achim Mumme.

**Writing – original draft:** Thomas Hummel.

**Writing – review & editing:** Dominic Mühlberger.

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
