## [Decision Letter · Decision Letter 0]

13 Oct 2020

PONE-D-20-25881

Perioperative changes of response to antiplatelet medication in common vascular surgery patients

PLOS ONE

Dear Dr. Hummel,

Thank you for submitting your manuscript to PLOS ONE. After careful consideration, we feel that it has merit but does not fully meet PLOS ONE’s publication criteria as it currently stands. Therefore, we invite you to submit a revised version of the manuscript that addresses the points raised during the review process.

The reviewers have commented on your above paper. They have suggested that this manuscript be revised according to the reviewers suggestions and resubmitted.  Provided you address the changes recommended, the manuscript will be reconsidered. 

We look forward to receiving your revised manuscript.

Kind regards,

Prof. Raffaele Serra, M.D., Ph.D

Academic Editor

PLOS ONE

Journal Requirements:

2. In your Methods section, please provide additional information about the participant recruitment method and the demographic details of your participants. Please ensure you have provided sufficient details to replicate the analyses such as: a) the recruitment date range (month and year), b) a description of how participants were recruited, and c) descriptions of where participants were recruited and where the research took place.

3. Please provide a sample size and power calculation in the Methods, or discuss the reasons for not performing one before study initiation.

4. Please include additional information regarding the survey or questionnaire used in the study and ensure that you have provided sufficient details that others could replicate the analyses. For instance, if you developed a questionnaire as part of this study and it is not under a copyright more restrictive than CC-BY, please include a copy, in both the original language and English, as Supporting Information.

5. Please ensure you have thoroughly discussed any potential limitations of this study within the Discussion section, including the potential impact of confounding factors.

Additional Editor Comments (if provided):

The reviewers have commented on your above paper. They have suggested that this manuscript be revised according to the reviewers suggestions and resubmitted.

Reviewers' comments:

Reviewer's Responses to Questions

**Comments to the Author**

1. Is the manuscript technically sound, and do the data support the conclusions?

Reviewer #1: Yes

Reviewer #2: Partly

2. Has the statistical analysis been performed appropriately and rigorously? 

Reviewer #1: Yes

Reviewer #2: No

3. Have the authors made all data underlying the findings in their manuscript fully available?

Reviewer #1: Yes

Reviewer #2: Yes

4. Is the manuscript presented in an intelligible fashion and written in standard English?

Reviewer #1: Yes

Reviewer #2: Yes

5. Review Comments to the Author

Reviewer #1: 1- Abstract, page 2, lines 34-35: The sentence describing high or low response for platelet activity is awkward and difficult to understand- please revise/clarify further.

2- Abstract, page 2, lines 52-54: There is a comment about ALR increasing, did CLR change?

3- Abstract, page 2, line 51- diabetes and insulin are confounded by indication-- they must be somewhat co-linear and should likely not be reported separately as independent risk factors.

4- Introduction, page 3, lines 65-78: It would be helpful if the authors could cite peer reviewed evidence about treatment failures that are ascribed to failure in antiplatelet therapy- can they provide prevalence information that is linked to actual clinical events/outcomes? This would further highlight clinical (and not experimental) significance and importance of studying platelet activity in vascular surgery patients.

5- Methods, page 4, lines 97-102: for the described exclusion criteria- please define renal impairment. Clarify 'recent' GI bleeding history. What is 'severe' cardiac disease?

6- Methods- Inclusion/Exclusion criteria- please present a flow diagram demonstrating the total denominator and ultimate numerator of patients included/excluded.

7- Methods- how were patients selected- was it only those seen in the clinic? Was there a group who undergo surgery vs group not undergoing surgery - may help for trying to establish baseline comparisons [controls]- please clarify.

8- Results, page 14- title of Figure 2 has a typographical error.

9- Results- page 15-16: The authors need to describe the types of operations that the patients underwent and differentiate platelet activity based upon the types of procedures.

10- Results- is there any clinical outcome information- adverse thromboembolic complications that occurred in the patients?

11- Discussion- can the authors provide a deeper discussion surrounding how if validated, this information will be used for point of care testing and clinical decision making?

Reviewer #2: This prospective observational study examines the responsiveness to aspirin and clopidogrel in patients before and after vascular surgery procedures. 176 patients were included, about 85% of whom were on dual antiplatelet therapy, the rest on one drug or the other, and multiple electrode aggregometer (Multiplate) was used to assess platelet responsiveness to the medications. The main findings are that 13% of patients were low responders to aspirin and 32% low responders to clopidogrel preoperatively, and low responsiveness to aspirin increased to 27.5% post-procedure.

This study seems relatively important and clinically significant, because as the authors point out studies like this have focused almost exclusively on cardiology patients, not on vascular or other surgery patients. There are some concerns that should be addressed, some of them substantial, that are concerning with this study and these are outlined below.

The authors should put forth a hypothesis being tested, otherwise it looks more like the authors are just looking for statistically significant relationships among variables.

The abstract, and maybe even the title should specify the study design as prospective observational (I think that it correct).

The abstract is not written clearly. The first sentence is the most problematic with the “/” and the 4 abbreviations all in one sentence. The authors should simply tell us what the issues are and why this study is being done before throwing a confusing sentence at the reader. A hypothesis should be included in the abstract - at the end of the introduction. Page 2 – LM 43 – risk factors for what? Page 2 – LM 48 – add “ taking antiplatelet medications aspirin and/or clopidogrel”. Almost all good papers have at least one P value in the abstract and this paper does not.

Page 3 – LM 73 – Several times the authors use the term “common vascular patients”. This should read “undergoing common vascular procedures”. A procedure can be common, but a patient cannot.

Page 3 – LM 87 – Here it is fine to say “common vascular procedure”, however in the methods section the authors need to tell the reader what these procedures are. Angioplasty, angiogram, stent placements, open surgery, endovascular surgery, etc.

Page 5 – LM 133 – Is it clearly evident to use these specific cutoffs to dichotomize patients into responders and non-responders?

Are there interactions between the response to aspirin and clopidogrel that would confound the results of this study? Since 84% of patients were on both, can we truly be sure the aspirin effect is not confounded by the other drug?

Table 1 – The rule of significant digits should be followed. Unless age was collected to the precision of 0.01 years, then the mean and SD do not get reported to this degree of precision. See the New England Journal for example, where almost all papers report age in whole numbers – no decimal places. Percentages are usually reported to the 0.1 degree of precision. P values are not given to 4 places after the decimal point in good journals. Usually 2 significant digits is the practice.

Does it matter if patients were on enteric coated vs. regular aspirin? If so, this should be added.

One major concern is that there were 30 variables assessed for their relationship to platelet responsiveness, but no post hoc adjustment for multiple comparisons (e.g. Bonferroni). Taking this into account, a P value of 0.00167 would be required for significance, otherwise we will see false positive findings. 2 of 3 findings in table 2, and 3 of 4 findings in table 3 would thus change to being not significant.

Page 8 – LM 192 – This finding really should have a P value as it seems like one of the main findings in the paper.

Page 8 – Several times we see “p<0.05” but the text should give the actual p value, especially given the issue with multiple comparisons mentioned above.

Page 11 – LM 253-255 – This sentence should be part of a limitations paragraph, which, by the way, is missing and should be added.

Discussion – It would be most interesting to expand on proposed mechanisms for the decreased responsiveness to anti PLT medications after the procedures.

Verify Now is no longer from Accumetrics, it is now from Instrumentation Laboratories. Add Bedford, Massachusetts as well.

In the conclusion, it would be more impactful and clear to say the percentage of patients who were low responders approximately doubled postoperatively, or say there was an approximate 2-fold increase (13.13% to 27.5%).

I don’t see a figure legend. Is figure 1 showing preop or postop findings? A legend is needed.

Figure 2 – same thing. “pot-OP” should be “post-OP”. Numbers should have one digit after the decimal point.

6. PLOS authors have the option to publish the peer review history of their article (what does this mean?). If published, this will include your full peer review and any attached files.

Reviewer #1: No

Reviewer #2: **Yes: **Steven M. Frank MD

---

## [Author Response · Author response to Decision Letter 0]

14 Nov 2020

Response to the academic editor´s and reviewers comments

PONE-D-20-25881

Perioperative changes of response to antiplatelet medication in vascular surgery patients

Dear Prof. Raffaele Serra, 

Thank you very much for the detailed reviewer’s critique of our manuscript new entitled „Perioperative changes of response to antiplatelet medication in vascular surgery patients”. We have reformatted our manuscript according to the PLOS ONE's style and reference style requirements. Furthermore we provide additional information about the participant recruitment and a sample size and power calculation in the methods section. Moreover, added a post hoc adjustment for multiple comparisons (Holm-Bonferroni method) and discuss major limitations of this study in the end of the manuscript. Please find attached our revised manuscript and our counterstatement to the reviewers’ comments.

We hope to have addressed the reviewers’ comments in an adequate manner and deeply hope that this substantially revised manuscript will be considered for publication in PLOS ONE.

Sincerely, yours

Thomas Hummel

Response to the comments of the academic editor:

PLOS ONE´s style requirements are added and carefully implemented in the manuscript.

2. In your Methods section, please provide additional information about the participant recruitment method and the demographic details of your participants. Please ensure you have provided sufficient details to replicate the analyses such as: a) the recruitment date range (month and year), b) a description of how participants were recruited, and c) descriptions of where participants were recruited and where the research took place.

The recruitment of the participants at the time of inpatient admission took place at the department of vascular surgery of the St. Josef Hospital, university hospital of the Ruhr-University of Bochum from February 2010 to December 2011. The study was designed as a prospective observational study and part of a medical doctoral thesis of the Ruhr-University of Bochum. The time of the patient´s selection was associated to the graduator and his temporary presence in the clinic. Therefore the recruitment time period was prolonged and the time for statistical analysis was also extended. Nevertheless we believe that our data are consistent and important to understand periprocedural changes in response of antiplatelet medication in vascular surgery patients. 

We added recruitment information, demographic detail and date range of the study to the Methods section (page: 3+4, lines 91-110)

3. Please provide a sample size and power calculation in the Methods, or discuss the reasons for not performing one before study initiation.

The sample size planning was performed using a significance level of α=0,05, estimated Pearson´s correlation coefficient of r=0,25 and stochastic power of 0,8. The sample size of the study calculated in this way was n=123 in order to get a significant result. The sample size and power calculation is added in the Methods. (page: 6, lines 165-168)

4. Please include additional information regarding the survey or questionnaire used in the study and ensure that you have provided sufficient details that others could replicate the analyses. For instance, if you developed a questionnaire as part of this study and it is not under a copyright more restrictive than CC-BY, please include a copy, in both the original language and English, as Supporting Information.

There was no questionnaire used in the study. All additional information regarding the survey is accessible in the publication.

5. Please ensure you have thoroughly discussed any potential limitations of this study within the Discussion section, including the potential impact of confounding factors.

We added limitations of our study in the Discussion section (page: 14+15, line: 375-405)

6. PLOS requires an ORCID iD for the corresponding author in Editorial Manager on papers submitted after December 6th, 2016.

The ORCID iD for the corresponding author (ORCID iD: https://orcid.org/0000-0001-6931-5930) is added in the Editorial Manager.

Response to the comments of reviewer #1:

1- Abstract, page 2, lines 34-35: The sentence describing high or low response for platelet activity is awkward and difficult to understand- please revise/clarify further.

This sentence is clarified and the abbreviations are explained later on in the manuscript. (page: 2, line: 34-35)

2- Abstract, page 2, lines 52-54: There is a comment about ALR increasing, did CLR change?

Results of CLR were added. (page: 2, line: 54-55)

3- Abstract, page 2, line 51- diabetes and insulin are confounded by indication—they must be somewhat co-linear and should likely not be reported separately as independent risk factors.

These are related factors, so they are connected in the abstract, also post hoc adjustment changes significance level and results (page: 2, line52-53). 

4- Introduction, page 3, lines 65-78: It would be helpful if the authors could cite peer reviewed evidence about treatment failures that are ascribed to failure in antiplatelet therapy- can they provide prevalence information that is linked to actual clinical events/outcomes? This would further highlight clinical (and not experimental) significance and importance of studying platelet activity in vascular surgery patients.

The Introduction section has been revised and the useful, but small, amount of evidence regarding surveillance and outcome was added. (page: 3, line: 71-76)

5- Methods, page 4, lines 97-102: for the described exclusion criteria- please define renal impairment. Clarify ‘recent’ GI bleeding history. What is ‘severe’ cardiac disease?

Reasonable and understandable objection to the described study exclusion criteria. Thank you for that. This prospective study was designed as a observational study, so that the exclusion criteria were limited to general contraindications to participation in a scientific study (as cognitive disorders, pregnancy…), and contraindications to the study medication or intrinsic activation / or depression of platelet activity (Gastrointestinal ulceration with history of bleeding within six weeks, NYHA stage 4, kidney or liver failure, abnormal platelet count, known coagulation disorders). The exclusion criteria have been newly explained and specified in the manuscript. (page : 5, line: 127-133)

6- Methods- Inclusion/Exclusion criteria- please present a flow diagram demonstrating the total denominator and ultimate numerator of patients included/excluded.

To our regret, we cannot present a flowchart for patient inclusion and exclusion. The study was designed as an observational study. Moreover, patients were recruited in the daily clinical routine. Therefore we have not detected all other patients of our department. (See response to Point 5 and 7)

7- Methods- how were patients selected- was it only those seen in the clinic? Was there a group who undergo surgery vs group not undergoing surgery - may help for trying to establish baseline comparisons [controls]- please clarify.

The recruitment of the patients was at the time of inpatient admission and was designed as a observational study and part of a graduation as a medical doctor of the Ruhr-University of Bochum. Therefore the recruitment was discontinuous, as it is unfortunately common in this setting. There was no planed control group in this study. Each participant had a symptomatic vascular disease with an indication for vascular treatment with different degrees of invasiveness (endovascular / local TEA / peripheral bypass). We hypothesized that low response is increased after vascular therapy in common vascular procedures, with regard to the invasiveness of the procedure. The recruitment information was added in the methods section. (page: 3+4; line: 91-95)

8- Results, page 14- title of Figure 2 has a typographical error.

The typographical error has been corrected. (Fig 2). We apologize for this.

9- Results- page 15-16: The authors need to describe the types of operations that the patients underwent and differentiate platelet activity based upon the types of procedures.

The changes of response to ASA in the different groups of operations are added (page: 9+10, line: 248-258)

Procedures are now described in the methods section (page: 4, line 103-109)

10- Results- is there any clinical outcome information- adverse thromboembolic complications that occurred in the patients?

We totally agree with reviewer #1. Clinical outcome information or the detection of thromboembolic complications would be very interesting to evaluate the impact of ALR /CLR in patients outcome after different vascular procedures. Unfortunately, no outcome or follow up data were collected from the patients. The aim of our study was the detection of changes in antiplatelet response in perioperative comparison. We believe this study could be the basis for further investigations with clinical endpoints and follow up platelet response measurements. Moreover we discuss this interesting issue in the limiations section. (page15, line 393-405)

11- Discussion- can the authors provide a deeper discussion surrounding how if validated, this information will be used for point of care testing and clinical decision making?

The impact of our periprocedural findings in clinical context and our daily practice of testing is explained at the end of the Discussion section. (page: 15, line: 405-409)

Response to the comments of Prof. Dr. Steven M. Frank / reviewer #2:

The authors should put forth a hypothesis being tested, otherwise it looks more like the authors are just looking for statistically significant relationships among variables.

We formulated our hypothesis at the end of the introduction section. (page: 3, line: 84-86)

The abstract, and maybe even the title should specify the study design as prospective observational (I think that it correct).

We specified the study design in the introduction section of the abstract. (page: 2, line: 38)

The abstract is not written clearly. The first sentence is the most problematic with the “/” and the 4 abbreviations all in one sentence. The authors should simply tell us what the issues are and why this study is being done before throwing a confusing sentence at the reader. A hypothesis should be included in the abstract - at the end of the introduction. Page 2 – LM 43 – risk factors for what? Page 2 – LM 48 – add “ taking antiplatelet medications aspirin and/or clopidogrel”. Almost all good papers have at least one P value in the abstract and this paper does not.

The first sentence of the abstract is clarified and the abbreviations are explained later on in the manuscript. (page: 2, line: 34-35). We formulated the hypothesis at the end of the introduction of the abstract (page: 2, line: 38-39) and specified the risk factors (page: 2, line: 43-44). At page 2 line 49 we added “ taking antiplatelet medications aspirin and/or clopidogrel”. P value was added for changes in response. (page: 2, line: 54).

Page 3 – LM 73 – Several times the authors use the term “common vascular patients”. This should read “undergoing common vascular procedures”. A procedure can be common, but a patient cannot.

The term “common vascular patient” was changed in the whole manuscript and even in the title.

Page 3 – LM 87 – Here it is fine to say “common vascular procedure”, however in the methods section the authors need to tell the reader what these procedures are. Angioplasty, angiogram, stent placements, open surgery, endovascular surgery, etc.

The performed procedures are explained in the methods section. (page: 4, lane: 103-109)

Page 5 – LM 133 – Is it clearly evident to use these specific cutoffs to dichotomize patients into responders and non-responders?

The cut off values used for the detection of ALR and CLR are part of the Multiplate instruction for use. The cut-off value >46 U of the ADPtest for CLR is well established and underlaid with good evidence and clinical outcome measurement based on the outlined reference (Sibbing D. et al. n=1608 cardiology patients undergoing intervention with increased risk in comparison to normal responders). In the detection of ALR different cut-off values are recommended by the manufacturer and used in the literature (Roche Diagnostics Multiplate). For the detection of a strong inhibition of cyclooxygenase (COX) -1 by ASA is recommended <30 U as cut-off (based on a study with n=76 by Pape et al. 2007). For the detection of an adequate inhibition of COX-1 by ASA <40 U is recommended as a cut-off. These cut off is based on the results of the stated study in the manuscript by Al-Azzam et al. with n=418 participants. However, no information is given on possible areas of application of the different cut-off values for ALR and there are no clinical outcome studies available based on the different cut off values with the range from 30 to 40 U for ASA used in the literature. In the present study, the ASA response was classified using the cut-off <40 U. The lack of evidence and different cut-off values used in the literature in the detection of ALR using Multiplate represents a limitation of the study, which was added in the limitations section at the end of the discussion (page: 14, line: 375-381).

Are there interactions between the response to aspirin and clopidogrel that would confound the results of this study? Since 84% of patients were on both, can we truly be sure the aspirin effect is not confounded by the other drug?

There were 148 (84.1%) patients on medication with 100 mg aspirin, 12 (6.8%) on dual antiplatelet medication with 100 mg aspirin and 75 mg clopidogrel, 11 (6.3%) on monotherapy with 75 mg clopidogrel, two (1.1%) on dual antiplatelet medication with 300 mg aspirin and 75 mg clopidogrel and 3 (1.7%) with 300 mg aspirin included in this study. Because of the small number of patients on dual medication, we can rule out any relevant influence on the presented results

Table 1 – The rule of significant digits should be followed. Unless age was collected to the precision of 0.01 years, then the mean and SD do not get reported to this degree of precision. See the New England Journal for example, where almost all papers report age in whole numbers – no decimal places. Percentages are usually reported to the 0.1 degree of precision. P values are not given to 4 places after the decimal point in good journals. Usually 2 significant digits is the practice.

Tables were revised and in the text accordingly corrected.

Does it matter if patients were on enteric coated vs. regular aspirin? If so, this should be added.

There is little evidence, that enteric coated ASA and regular aspirin as clopidogrel hydrogen sulfate and clopidogrel besylate could have different impaired antiplatelet effect. In our study we had a homogenous medication with regular aspirin and clopidogrel hydrogen sulfate. Therefore we only used aspirin nomenclature in the text. The confusing enteric labeling of the tables has been changed. (see titles table 1 & 2).

One major concern is that there were 30 variables assessed for their relationship to platelet responsiveness, but no post hoc adjustment for multiple comparisons (e.g. Bonferroni). Taking this into account, a P value of 0.00167 would be required for significance, otherwise we will see false positive findings. 2 of 3 findings in table 2, and 3 of 4 findings in table 3 would thus change to being not significant.

We added the post hoc adjustment for multiple comparisons (Holm-Bonferroni method). Therefore initial findings in the risk factor analysis changes and were added in the results section. Thus the discussion was adapted to the findings in the post hoc analysis. (page : 7, line: 204-209) (page: 10, line: 252-254) (page: 10, line: 255-2262)

Page 8 – LM 192 – This finding really should have a P value as it seems like one of the main findings in the paper.

Good and reasonable point. Thanks for that. P value was added to this main finding. (page: 9, line: 224-225)

Page 8 – Several times we see “p<0.05” but the text should give the actual p value, especially given the issue with multiple comparisons mentioned above.

The actual p values with addition of post hoc adjustment for multiple comparisons have been added in the manuscript. 

Page 11 – LM 253-255 – This sentence should be part of a limitations paragraph, which, by the way, is missing and should be added.

We added limitations of our study in the discussion section. (page: 14+15, line: 375-404)

Discussion – It would be most interesting to expand on proposed mechanisms for the decreased responsiveness to anti PLT medications after the procedures.

The discussion was changed, expanded and rearranged according to the post hoc analysis. (page: 13, line: 329-331) (page: 14, line347-361)

Verify Now is no longer from Accumetrics, it is now from Instrumentation Laboratories. Add Bedford, Massachusetts as well.

Correct. We are sorry. Actual information for VerifyNow is added. (page: 13, line: 322)

In the conclusion, it would be more impactful and clear to say the percentage of patients who were low responders approximately doubled postoperatively, or say there was an approximate 2-fold increase (13.13% to 27.5%).

We added the more impactful statement in the conclusion. (page: 16, line413-415)

I don’t see a figure legend. Is figure 1 showing preop or postop findings? A legend is needed.

Figure legends have been added.

Figure 2 – same thing. “pot-OP” should be “post-OP”. Numbers should have one digit after the decimal point.

Figure 2 has been corrected.

---

## [Decision Letter · Decision Letter 1]

1 Dec 2020

PONE-D-20-25881R1

Perioperative changes of response to antiplatelet medication in vascular surgery patients

PLOS ONE

Dear Dr. Hummel,

Thank you for submitting your manuscript to PLOS ONE. After careful consideration, we feel that it has merit but does not fully meet PLOS ONE’s publication criteria as it currently stands. Therefore, we invite you to submit a revised version of the manuscript that addresses the points raised during the review process.

The manuscript was substantially improved but some minor revisions are still needed before final acceptance.

We look forward to receiving your revised manuscript.

Kind regards,

Prof. Raffaele Serra, M.D., Ph.D

Academic Editor

PLOS ONE

Reviewers' comments:

Reviewer's Responses to Questions

**Comments to the Author**

1. If the authors have adequately addressed your comments raised in a previous round of review and you feel that this manuscript is now acceptable for publication, you may indicate that here to bypass the “Comments to the Author” section, enter your conflict of interest statement in the “Confidential to Editor” section, and submit your "Accept" recommendation.

Reviewer #1: All comments have been addressed

Reviewer #2: (No Response)

2. Is the manuscript technically sound, and do the data support the conclusions?

Reviewer #1: Yes

Reviewer #2: Yes

3. Has the statistical analysis been performed appropriately and rigorously? 

Reviewer #1: Yes

Reviewer #2: Yes

4. Have the authors made all data underlying the findings in their manuscript fully available?

Reviewer #1: Yes

Reviewer #2: Yes

5. Is the manuscript presented in an intelligible fashion and written in standard English?

Reviewer #1: Yes

Reviewer #2: Yes

6. Review Comments to the Author

Reviewer #1: 1- Abstract, Introduction: 'Reduced antiplatelet activity of aspirin' is abbreviated as 'ALR' however in the final sentence of the introduction to the abstract, the authors have 'ARL'.

2- Abstract, Results- 2nd sentence highlights 'The prevalence at the time of admission'-- what prevalence are they referring too exactly- is it any evidence of reduced antiplatelet activity or was there a specific threshold from the assay that determined a 'yes' or 'no' to the presence of reduced antiplatelet activity. Please clarify.

3- Abstract, results line 52-53: the syntax/presentation of the sentence is awkward- please consider revising to clarify the results to 'The overall ALR increased significantly postoperatively to 27.5% (p=0.0006); however, there was no significant change in CLR that was detected.'

4- Discussion, line 273- the authors site a ALR rate of 13.1% but CLR rate of 20%; however, in the abstract, the CLR rate is 32% (line 50)- please clarify.

5- Discussion- line 273-274- the comment that the patients were 'already without effective protection against thromboembolic events before the necessary vascular intervention'-- I think I understand what the authors are intimating but without linkage clinically (either through peer reviewed literature- which the authors highlight is largely missing in the literature, or by evidence in the current study- which we don't have enough clinical events )- can you actually make this statement? While hypothetically at higher risk, without actual peer reviewed evidence and/or clinical end-points in the current analysis, can this statement be scientifically supported?

6- Discussion lines 320-322: what is the difference between 'LR' and 'ALR'?

7- Given the prevalence of aspirin/clopidogrel resistance- do the authors propose routine platelet aggregometry assays for all vascular patients undergoing intervention? It is unclear what without direct linkage to major adverse clinical events how to use the current findings in clinical practice. I think the observation is notable and I agree with the authors that additional study is warranted.

8- Can the authors provide an explanation as to why aspirin resistance increased postoperatively but Plavix did not?

Reviewer #2: Thank you for the opportunity to review this revised manuscript from Hummel et al., which aimed to determine the prevalence of platelet reactivity for aspirin and clopidogrel before and after common vascular procedures. Overall, the manuscript is significantly improved from the initial submission. Please find below a few lingering concerns that can help improve the manuscript.

Abstract L51-53: P-values should be included in the abstract when describing risk factors for ALR (insulin and CRP).

Introduction L90-92: The phrase “with regard to the invasiveness of the procedure” is confusing in the hypothesis. Are the authors hypothesizing that more invasive procedures will lead to a higher incidence of low response rate? Please clarify.

Methods L96-101: These are results (demographic and clinical characteristics) and should be moved to the Results section of the manuscript.

Discussion L443-451: This paragraph on the clinical significance of the results and determining who should receive these tests is, in my opinion, a very important point that needs to be expanded upon in the Discussion. In reading the manuscript, a big question that remains is, “what does this mean for patients?” While the 2-fold increase in low responders to aspirin is striking, it is important to discuss this in terms of what it means for patient care and clinical outcomes. Would you consider the patients you enrolled in this study “high risk,” and thus justified in receiving these tests? I encourage the authors to further address these points.

Table 3: What does the row “Operative Procedure” under the Treatment subcategory refer to in this table? This is not part of the 5 groups that were discussed in the Methods section for operative approach. From the manuscript, it seems like this should be rephrased to “Open Operative Approach.”

7. PLOS authors have the option to publish the peer review history of their article (what does this mean?). If published, this will include your full peer review and any attached files.

Reviewer #1: No

Reviewer #2: **Yes: **Steven M. Frank

---

## [Author Response · Author response to Decision Letter 1]

6 Dec 2020

Response to the comments of reviewer #1:

1- Abstract, Introduction: 'Reduced antiplatelet activity of aspirin' is abbreviated as 'ALR' however in the final sentence of the introduction to the abstract, the authors have 'ARL'.

We changed and clarified to the explained abbreviations in the final sentence of the introduction section of the abstract. (page: 2; line: 38)

2- Abstract, Results- 2nd sentence highlights 'The prevalence at the time of admission'-- what prevalence are they referring too exactly- is it any evidence of reduced antiplatelet activity or was there a specific threshold from the assay that determined a 'yes' or 'no' to the presence of reduced antiplatelet activity. Please clarify.

Important point. We clarified this sentence in a short way for the abstract (page: 2, line 48+49). Specific threshold and cut off values for the Multiplate aggregometry were explained in the materials and methods section in the manuscript (page: 6, line: 154-156).

3- Abstract, results line 52-53: the syntax/presentation of the sentence is awkward- please consider revising to clarify the results to 'The overall ALR increased significantly postoperatively to 27.5% (p=0.0006); however, there was no significant change in CLR that was detected.'

We changed and clarified this sentence. (page: 2; line: 51+52)

4- Discussion, line 273- the authors site a ALR rate of 13.1% but CLR rate of 20%; however, in the abstract, the CLR rate is 32% (line 50)- please clarify.

Astonishing that despite extensive intern text revision, such obvious errors are still overlooked. We are very sorry and glad that we have the opportunity to revise this mistake in the discussion section. Thank you for that. CLR rate was corrected to 32% and discussion was modulated. (page: 11; line: 268 + page: 21; line: 276-279))

5- Discussion- line 273-274- the comment that the patients were 'already without effective protection against thromboembolic events before the necessary vascular intervention'-- I think I understand what the authors are intimating but without linkage clinically (either through peer reviewed literature- which the authors highlight is largely missing in the literature, or by evidence in the current study- which we don't have enough clinical events )- can you actually make this statement? While hypothetically at higher risk, without actual peer reviewed evidence and/or clinical end-points in the current analysis, can this statement be scientifically supported?

Reasonable and understandable objection to the clinical impact of low response. We agree that the induced clinical outcome association of low response is inappropriate at this point of the text. The sentence was defused and corrected. (page: 11; line: 267-270). Later on in the discussion section, the evidence of the clinical impact of a low response is discussed. (page: 14, line 348-359)

6- Discussion lines 320-322: what is the difference between 'LR' and 'ALR'?

We clarified and changed LR (any low response to antiplatelet medication) to ALR (aspirin low response). (page: 13; line: 315-318)

7- Given the prevalence of aspirin/clopidogrel resistance- do the authors propose routine platelet aggregometry assays for all vascular patients undergoing intervention? It is unclear what without direct linkage to major adverse clinical events how to use the current findings in clinical practice. I think the observation is notable and I agree with the authors that additional study is warranted.

We totally agree to this objection. At the end of the discussion, we comment on this topic and also disclose our behavior in our daily practice. (page: 15+16, line: 389-404)

8- Can the authors provide an explanation as to why aspirin resistance increased postoperatively but Plavix did not?

Unfortunately, the informative value of the study with regard to clopidogrel is limited due to the small number of patients in this group. This point was taken into account in the evaluation of the results, discussion and the study limitations (page: 15, line: 385-388 + page: 12, line: 276-279).

Response to the comments of Prof. Dr. Steven M. Frank / reviewer #2:

Abstract L51-53: P-values should be included in the abstract when describing risk factors for ALR (insulin and CRP).

P-values were added to the results section of the abstract. (page: 2; line: 50+51)

Introduction L90-92: The phrase “with regard to the invasiveness of the procedure” is confusing in the hypothesis. Are the authors hypothesizing that more invasive procedures will lead to a higher incidence of low response rate? Please clarify.

We clarified the hypothesis at the end of the introduction section. (page: 3; line:80-82)

Methods L96-101: These are results (demographic and clinical characteristics) and should be moved to the Results section of the manuscript.

These patients characteristics were moved to the beginning of the results section. (page: 6; line: 170-173)

Discussion L443-451: This paragraph on the clinical significance of the results and determining who should receive these tests is, in my opinion, a very important point that needs to be expanded upon in the Discussion. In reading the manuscript, a big question that remains is, “what does this mean for patients?” While the 2-fold increase in low responders to aspirin is striking, it is important to discuss this in terms of what it means for patient care and clinical outcomes. Would you consider the patients you enrolled in this study “high risk,” and thus justified in receiving these tests? I encourage the authors to further address these points.

Important note, we expanded the end of the discussion section and clarified our point of view in terms of patient´s testing. (page: 15, line: 389-404)

Table 3: What does the row “Operative Procedure” under the Treatment subcategory refer to in this table? This is not part of the 5 groups that were discussed in the Methods section for operative approach. From the manuscript, it seems like this should be rephrased to “Open Operative Approach.”

We corrected table 3 and rephrased treatment subcategory. (table 3)

---

## [Editor Report · Decision Letter 2]

8 Dec 2020

Perioperative changes of response to antiplatelet medication in vascular surgery patients

PONE-D-20-25881R2

Dear Dr. Hummel,

We’re pleased to inform you that your manuscript has been judged scientifically suitable for publication and will be formally accepted for publication once it meets all outstanding technical requirements.

Kind regards,

Prof. Raffaele Serra, M.D., Ph.D

Academic Editor

PLOS ONE

Additional Editor Comments (optional):

amended manuscript is acceptable.
---

## [Editor Report · Acceptance letter]

16 Dec 2020

PONE-D-20-25881R2 

Perioperative changes of response to antiplatelet medication in vascular surgery patients 

Dear Dr. Hummel:

I'm pleased to inform you that your manuscript has been deemed suitable for publication in PLOS ONE. Congratulations! Your manuscript is now with our production department. 

Kind regards, 

on behalf of

Prof. Raffaele Serra 

Academic Editor

PLOS ONE